# Supernumerary Head of the Biceps Brachii Muscle Influences the Topography of the Coracobrachialis and Biceps Brachii Muscles

**DOI:** 10.3390/medicina60111726

**Published:** 2024-10-22

**Authors:** Yu-Ran Heo, Hyunsu Lee, Si-Wook Lee, Beom-Soo Kim, Hong-Tae Kim, Jae-Ho Lee

**Affiliations:** 1Department of Anatomy, Keimyung University School of Medicine, Daegu 42601, Republic of Korea; yrheo@yuhs.ac; 2Department of Physiology, Pusan National University School of Medicine, Yangsan 50612, Republic of Korea; hyunsu.lee@pusan.ac.kr; 3Department of Orthopedic Surgery, Keimyung University Dongsan Hospital, Daegu 42601, Republic of Korea; shuk@dsmc.or.kr (S.-W.L.); bskim@dsmc.or.kr (B.-S.K.); 4Department of Anatomy, School of Medicine, Catholic University of Daegu, Daegu 42472, Republic of Korea; htaekim@cu.ac.kr

**Keywords:** biceps brachii, coracobrachialis, supernumerary head, anatomical variation

## Abstract

*Background/Objectives*: Anatomical variations in the biceps brachii muscle (BBM) are extremely frequent, leading to developmental and clinical implications. We studied the topography of the BBM and analyzed its correlations with other structures in the brachial region. *Methods:* A total of 103 cadaveric upper extremities were dissected. The length of the upper extremities was utilized as a reference line. The origin and insertion of the BBM, the coracobrachialis muscle (CBM), and the related neurovascular system were evaluated. *Results*: Each variable was calculated as a percentile and compared according to the presence of the considered variation; in particular, a supernumerary head of the BBM was found in 12/103 (11.65%) of upper extremities and was associated with a longer upper limb (506.25 ± 32.55 mm vs. 484.27 ± 30.41 mm, *p* = 0.022). When the variables were standardized by the length of the upper limb, the accessory head of the BBM was associated with the distal insertion point of the CBM (28.18 ± 3.54% vs. 30.59 ± 2.94%, *p* = 0.011) and BBM length (55.11 ± 2.17% vs. 58.18 ± 3.72%, *p* = 0.006). Other variables did not present significant differences with respect to the presence of the supernumerary head of the BBM. *Conclusions:* BBM variations may affect the topography of other structures, such as the length of the upper extremities, insertion of the CBM, and length of the BBM. Further studies are required to elucidate its clinical implications.

## 1. Introduction

In the human body, the anterior brachial region comprises many structures, including those associated with the muscular, nervous, and vascular systems. The muscles in this region include the coracobrachialis (CBM), biceps brachii (BBM), and brachialis muscles, which are responsible for elbow flexion. The neurovascular system of the anterior brachial region contains terminal branches of the brachial plexus, the brachial artery, and its branches. The terminal branches of the brachial plexus are the musculocutaneous, median, ulnar, radial, and axillary nerves. Among these, the nerves related to the anterior part of the brachial region are the median and the musculocutaneous nerves. The brachial artery is a continuation of the axillary artery, which supplies the anterior part of the brachial region.

In the anterior brachial region, the most powerful flexor muscle is the BBM, which is also a powerful supinator of the forearm [1]. Anatomically, the BBM has two heads, namely, short and long heads. The origin of the short head of the BBM is the coracoid process, while that of the long head is the supraglenoid tubercle of the scapula. Additionally, this muscle is inserted into the radial tuberosity and bicipital aponeurosis. The BBM is innervated by the MCN and supplied by the brachial artery. Traditionally, the BBM is described as a two-headed muscle; however, variations of the BBM, such as supernumerary heads, have been reported [2,3,4,5]. Previous studies have suggested that such BBM variations may be related to the variation of the MCN [6,7], as the penetration site of the MCN differs according to the presence of the supernumerary head of the BBM. However, studies on the correlations between the BBM and other structures of the anterior brachial region in the upper extremities remain rare.

In this study, the anterior brachial region in 103 upper extremities was dissected to recognize the topography of the BBM, CBM, and MCN, as well as to investigate the association between the supernumerary head of the BBM and other structures in the brachial region. The findings of this study have clinical implications and can be helpful for accurate diagnostic interpretation.

## 2. Materials and Methods

In this study, anterior brachial regions from 52 donated cadavers (103 cadaveric upper extremities) were dissected. Each cadaver was positioned in a supine position with the upper extremities extended. The skin and superficial fascia were removed by long deltopectoral incision and the deltopectoral fascia, pectoralis major, and deltoid muscles were identified. Then, the axillary sheath and adipose tissue were dissected to confirm the axillary and brachial arteries and the brachial plexus. The branches of the neurovascular structures were also identified (Figure 1). The median nerve is formed by the connection of the lateral and medial cords, and this point was named the root of the median nerve. The brachial region was dissected and the BBM, CBM, and MCN were identified. The CBM perforates the MCN and inserts into the middle of the medial aspect of the humerus. The distance between the coracoid process and proximal and distal insertions of the CBM, BBM insertion (short head length), the root of the median nerve, the point of the CBM pierced by the MCN, and the origin of the deep brachial artery (DBA) were measured. These variables were compared according to the presence of the supernumerary head of the BBM. Furthermore, the correlation between the variation and other muscles and structures was analyzed. The length of the upper extremities from the coracoid process to the end of third finger was taken as the reference line, and the length of each variable was standardized with respect to the reference line and represented as a percentile. The length of each structure was measured using digital calipers (NA500-300S, Blue bird, Yongin-si, Korea). Each measurement was performed twice with an accuracy of up to 0.1 mm by two anatomists.

All statistical analyses were performed using the Statistical Package for the Social Sciences (SPSS) (Ver. 24.0, IBM Armonk, New York, NY, USA). To compare the variables, the Mann–Whitney U-test was used. Statistical significance was defined as a two-tailed *p*-value < 0.05.

## 3. Results

The mean total length of the upper extremity (taken as the reference line) was found to be 486.83 ± 31.31 mm. The distances between the coracoid process and proximal and distal insertions of the CBM were 109.03 ± 16.78 mm and 147.36 ± 16.04 mm, respectively. The length of the short head of the BBM was 281.33 ± 23.97 mm. The origin of the DBA and the root of the MN were 76.5 ± 24.45 mm and 29.06 ± 24.81 mm, respectively. The point at which the MCN pierces the CBM was located at 48.43 ± 20.11 mm (Figure 2). Detailed information is presented in Appendix A. In addition, the detailed topographies of the upper structures, according to the presence or absence of the accessory head, are presented in Table 1. Its difference according to gender is summarized in Appendix A.

The supernumerary head of the BBM was observed in 11.65% (12/103) of upper extremities; moreover, four heads of BBM were found (Appendix A). The differences between upper structures, according to this variation, are summarized in Table 1. The accessory head was associated with a longer length of the upper limb (506.25 ± 32.55 mm vs. 484.27 ± 30.41 mm, *p* = 0.022). The point of the CBM pierced by the MCN also differed between these groups; however, this difference was not statistically significant (47.23 ± 20.1 mm vs. 57.5 ± 18.58 mm, *p* = 0.097). Other variables did not present any significant differences according to the presence of the supernumerary head.

The muscular and neurovascular topographies, according to the presence or absence of the supernumerary head, are presented as percentile data in Table 2. Considering the mean of the percentile with respect to the reference line, the distances between the coracoid process and proximal and distal insertions of the CBM were 22.38 ± 3.07 and 30.31 ± 3.03, respectively. The length of short head of the BBM was 57.82 ± 3.66. The origin of the DBA and the root of the MN were 15.77 ± 5.16 and 5.95 ± 5.30, respectively. The point of the CBM pierced by the MCN was at 9.88 ± 3.88.

The distal insertion of the CBM was located more proximally in arms with a supernumerary head (28.18 ± 3.54%) than in those without it (30.59 ± 2.94%, *p* = 0.011). The length of the short head of the BBM was shorter in arms with a supernumerary head (55.11 ± 2.17%) than in those without it (58.18 ± 3.72%, *p* = 0.006). The other structures did not vary according to the presence of the accessory head.

## 4. Discussion

The anterior compartment of the upper extremity contains the BBM, CBM, and brachialis muscles, and variations in the BBM and CBM are frequently observed. Muscles in the anterior brachial region of the upper extremity act to promote flexion of the elbow joint, and these muscles are innervated by the musculocutaneous nerve (MCN). The axillary and brachial arteries are responsible for blood supply to this region [8]. Variations in these neurovascular structures are also observed, and are often accompanied by variations in the abovementioned muscles. Therefore, the topography of neurovascular structures in the axilla and arm should be assessed according to major muscle variations.

First, we demonstrated the overall topography of the axilla and arm. From the coracoid process, the median nerve is formed from the lateral and medial cords at the 5 percentile level. Then, the MCN pierces the CBM at the 10 percentile level and the deep brachial artery originates from the brachial artery at the 15 percentile level. The CBM is widely inserted at the 20–30 percentile level of the upper limb. These data are similar to those reported in previous studies [6,9], and these major structures appear at regular intervals (about 5 percentile intervals). The landmarks of major anatomical structures at such regular intervals are clinically important and helpful. However, our results represent a relative value to the total length of upper limb and, thus, may differ from previous results [10,11], due to differences in the race or age of donated cadavers. A larger-scale study should be conducted using a clinically meaningful reference line.

Many authors have studied muscular and neurovascular variations in the upper extremities [2,3,12,13,14,15,16]. Of these variations, the number and pattern of the BBM heads are the most variable. The reported frequency of the accessory head of BBM ranges from 8% to 37.5% [17]. This variant muscle may cause idiopathic pain by neurovascular compression [18,19]. And it may be mistaken as a soft tissue tumor in ultrasound scans. Considering the high frequency and clinical importance of this variation, the topography of neurovascular structures was compared according to the presence of the accessory head. In this study, we found the supernumerary head of the BBM in 11.65% of cadavers. Interestingly, the upper limb was significantly longer in limbs with a supernumerary head than in those without it. The anterior compartment with a longer length requires more muscles; therefore, supernumerary muscles may have developed in response to the length. Embryologically, the mesoderm invades the upper limb bud and divides into ventral and dorsal muscular masses in the fifth week of development. The BBM is derived from the ventral mass, and its inappropriate cleavage may produce the supernumerary head. However, its embryological mechanism should be further investigated. Furthermore, the percentile level between the coracoid process and the coracobrachialis differed according to the variation of the BBM. In addition, the supernumerary head was associated with a shorter percentile of BBM length. This suggests that the total amount of muscle in the anterior part of the brachial region is fixed, such that the length of the original muscles can be shortened when additional muscles are present.

As a limitation in this study, no mention was made of differences by sex or age, as we sought to understand the overall correlation between the structures. Furthermore, when measuring the BBM, the short and long heads were not measured separately. Additionally, the length of the humerus is normally used as a reference line in research on the upper limb; however, in our study, the length of the entire upper limb was used as the reference line, given that the result was not statistically significant. As the upper limb length is not a clinically meaningful reference line, additional studies using other reference lines are needed. Considering that variations in this region are diverse, it is necessary to compare the topographies of these structures according to the pattern of the variations.

In this study, we demonstrated the topography of structures in the upper extremities and examined their correlations. It was demonstrated that variations in the upper extremities may affect the neuromuscular structures in this region. Differences in anatomical positions of neuromuscular structures due to the presence of a supernumerary head may cause confusion to clinicians.

## 5. Conclusions

Variations in the biceps brachii may affect the topography of other structures, such as the length of the upper extremities, insertion of the coracobrachialis, and length of the long head of the biceps brachii. Our results allowed for the determination of an embryological basis for the observed relationship between the muscular and neurovascular structures. Further studies should be carried out to explain and validate the possible clinical implications.

## Figures and Tables

**Figure 1 medicina-60-01726-f001:**
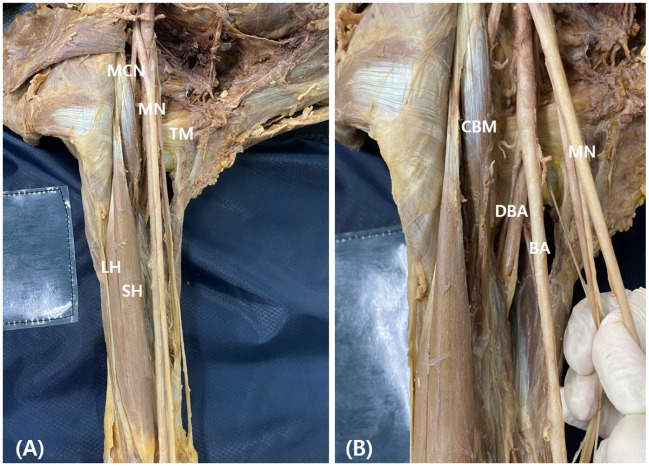
Representative photography of the muscular and neurovascular structures in arm (**A**) and upper arm (**B**). BA, brachial artery; CBM, coracobrachialis muscle; DBA, deep brachial artery. LH, long head; MCN, musculocutaneous nerve; MN, median nerve; SH, short head; TM, teres major muscle.

**Figure 2 medicina-60-01726-f002:**
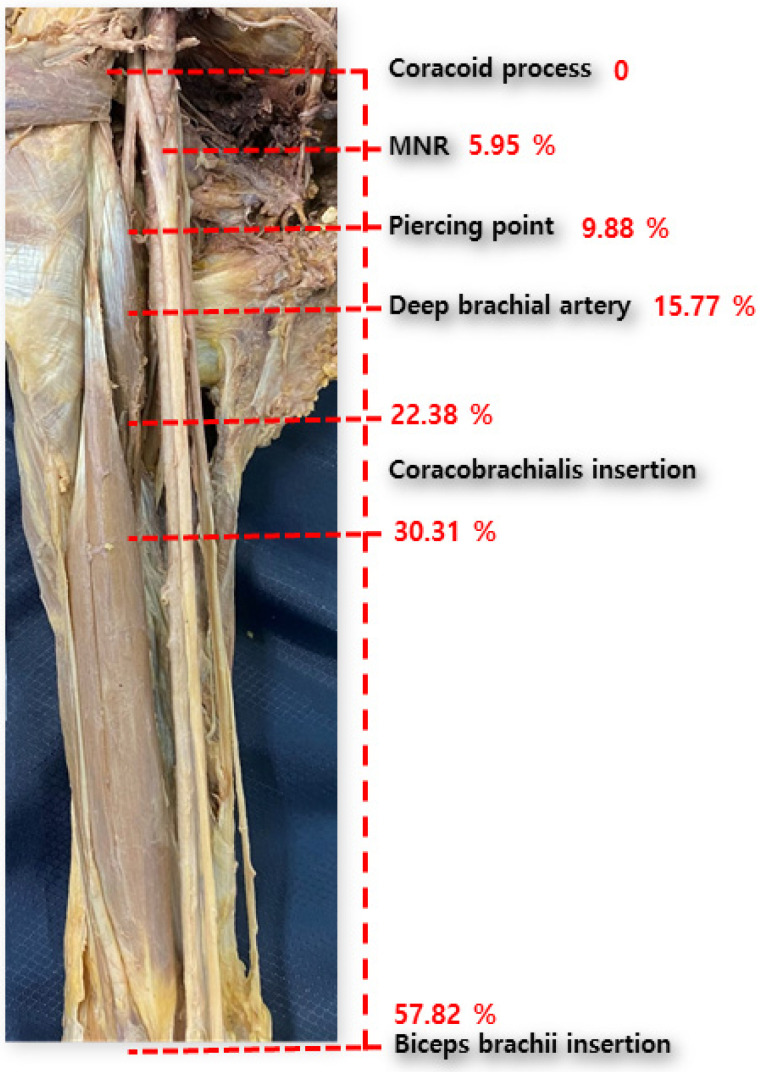
Schematic image of topography of muscular and neurovascular structures from the coracoid process.

**Table 1 medicina-60-01726-t001:** Topography of muscular and neurovascular structures according to the variation of the biceps brachii muscle.

		Supernumerary Head (mm)		
	Mean (*n* = 103)	(−) (*n* = 91)	(+) (*n* = 12)	*p*
Upper extremity	486.83 ± 31.31	484.27 ± 30.41	506.25 ± 32.55	0.022 *
CBM proximal insertion	109.03 ± 16.78	109.21 ± 17.03	107.67 ± 15.35	0.766
CBM distal insertion	147.36 ± 16.04	148.03 ± 15.94	142.25 ± 16.56	0.242
BBM (short head)	281.33 ± 23.97	281.67 ± 24.62	278.75 ± 17.77	0.692
DBA	76.5 ± 24.45	76.97 ± 23.71	72.92 ± 30.45	0.592
MNR	29.06 ± 24.81	27.82 ± 24.01	38.42 ± 29.73	0.166
Point of CBM pierced by MCN	48.43 ± 20.11	47.23 ± 20.1	57.5 ± 18.58	0.097

* *p* < 0.05; All locations were measured from the coracoid process, except for the short head of the biceps brachii muscle. BBM: biceps brachii; CBM: coracobrachialis muscle; DBA: deep brachial artery; MNR: median nerve root.

**Table 2 medicina-60-01726-t002:** Topography of muscular and neurovascular structures according to the variation of the biceps brachii muscle (as percentile).

		Supernumerary Head (%)		
	Mean (*n* = 103)	(−) (*n* = 91)	(+) (*n* = 12)	*p*
CBM proximal insertion	22.38 ± 3.07	22.52 ± 2.97	21.35 ± 3.42	0.213
CBM distal insertion	30.31 ± 3.03	30.59 ± 2.94	28.18 ± 3.54	0.011 *
BBM (short head)	57.82 ± 3.66	58.18 ± 3.72	55.11 ± 2.17	0.006 **
DBA	15.77 ± 5.16	15.93 ± 4.91	14.54 ± 6.2	0.374
MNR	5.95 ± 5.30	5.73 ± 5.05	7.57 ± 5.94	0.249
Point of CBM pierced by MCN	9.88 ± 3.88	9.69 ± 3.97	11.3 ± 3.37	0.185

* *p* < 0.05, ** *p* < 0.01; All locations were measured from the coracoid process, except the short head of the biceps brachii muscle. BBM: biceps brachii; CBM: coracobrachialis muscle; DBA: deep brachial artery; MNR: median nerve root.

## Data Availability

The original contributions presented in the study are included in the article. Further inquiries can be directed to the corresponding author.

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
