# Peer review of "Supernumerary Head of the Biceps Brachii Muscle Influences the Topography of the Coracobrachialis and Biceps Brachii Muscles"

_medicina, 2024, doi:10.3390/medicina60111726_

Round 1

Reviewer 1 Report

Comments and Suggestions for Authors

Next: these muscles are responsible for flexion - flexion of what? Besides, maybe short intro on embryology? Not necessary perhaps but sometimes it enriches the text and gains the readers attendance.

- What does it mean that terminal branches of the brachial plexus are ... here's a list of the nerves : isn't i.e.long thoracic nerve the nerve of brachial plexus? isn't it "terminal" for serratus anterior muscle?

- recently in my department we are finishing a huge study on the deep brachial artery: I do not agree with your statement  about the area it supplies...

- the whole intro should be rewritten

the discussion more focused on the topic of the study.

Study limitations - are these only limitations denoted?

Comments on the Quality of English Language

The text needs strong English editing - please use native speaker who should be a person knowing how to write in English

lots of grammatical errors, typos i.e. - line 44 - BBM has two head (heads maybe?)

Reviewer 2 Report

Comments and Suggestions for Authors

Dear authors,

It has been a pleasure to act as a reviewer of your manuscript ‘Supernumerary head of the biceps brachii muscle influences the topography of the coracobrachialis and biceps brachii muscles’. It is a useful and well-designed work and its results and conclusions are of great interest. However, the manuscript should be extensively improved before being considered for publication in any journal.

Introduction:

It is too vague and provides a lot of general information without dealing with the background of the particular subject. Lines 49-51 and 55-58 are not understandable.

Material and methods.

A supplementary table providing technical and demographical features of the sample should be included. Also, the dissection protocol is poorly explained.

Results

It is by far the best section of the manuscript. However, it needs further illustration, including a detailed photographical record of the 12 non-canonical cases. Anatomic figures should also be scaled and referenced according the cephalo-caudal and medio-lateral axis. In any case, finding are interesting and worthful.

Discussion.

The discussion is poor. It does not dwell into the potential clinical consequences (e.g, lines 135-136) of the findings and neglects recent research on the field (see Khan et al., 2020;  Szewczyk et al. 2022, 2023). The rationale of the information included in some paragraphs is difficult to understand without further development (e.g, lines 129-129). Also, when addressing limitations, they should be discussed and not merely listed, as it happens in lines 153-156. Some paragraphs are difficult to understand (e.g lies 126-127).

Overall, this is a well-designed study that poses an interesting question and provides an adequate answer, but scientific communication requires some ‘formal’ details that should not be overlooked. I do encourage authors to work more carefully in the manuscript, as I would like to see your results published.  As a personal comment: being myself not an English-speaker, I do profoundly believe that editors and reviewers who allow themselves to be picky regarding scientific English impose a barrier on science and contribute to transform academical publication into a highly biased business. However, a minimum degree of edition is mandatory, as incomplete or incoherent sentences lead to the blurring of real conclusions and might be interpreted as a lack of either respect for the readers or self-respect.

Un saludo.

Comments on the Quality of English Language

(see the general evaluation of the manuscript)